# Next-generation CRISPR gene-drive systems using Cas12a nuclease

Sara Sanz Juste [1,2], Emily M. Okamoto [3], Christina Nguyen[4], Xuechun Feng[3,5] ✉ & Víctor López Del Amo [4] ✉

One method for reducing the impact of vector-borne diseases is through the use of CRISPR-based gene drives, which manipulate insect populations due to their ability to rapidly propagate desired genetic traits into a target population. However, all current gene drives employ a Cas9 nuclease that is constitutively active, impeding our control over their propagation abilities and limiting the generation of alternative gene drive arrangements. Yet, other nucleases such as the temperature sensitive Cas12a have not been explored for gene drive designs in insects. To address this, we herein present a proof-of-concept gene-drive system driven by Cas12a that can be regulated via temperature modulation. Furthermore, we combined Cas9 and Cas12a to build double gene drives capable of simultaneously spreading two independent engineered alleles. The development of Cas12a-mediated gene drives provides an innovative option for designing next-generation vector control strategies to combat disease vectors and agricultural pests.

One strategy for controlling vector-borne pathogens and agricultural pests involves the use of CRISPR gene-drive systems, which represent a genetic tool for propagating desired traits into wild insect populations[1]. The engineered genetic elements that comprise gene drives are rapidly disseminated through a population because they are self-propagating and can bias their inheritance rate. Current CRISPR-based gene drives consist of a three-component transgene: (i) Cas9, a nuclease that produces DNA double-strand breaks; (ii) a guide RNA (gRNA) that leads Cas9 to cleave the DNA at a target site; and (iii) two homology arms matching both sides of the cut site to promote homology-directed repair (HDR). Gene drives propagate by first generating a DNA double-strand break in the wildtype allele, which is repaired by HDR using the gene-drive containing chromosome as a template, ultimately resulting in the original wildtype allele being replaced by that of a gene-drive one[2]. At the population level, when a gene-drive-engineered individual encounters a wildtype one, the subsequent allelic conversion promotes biased Mendelian inheritance (50%) towards super-Mendelian inheritance (>50%), allowing rapid gene drive propagation through the

population. In fact, mosquitoes with CRISPR gene-drive modifications have been observed to reach nearly 100% inheritance of the desired allele in consecutive generations[3–6].

However, even though these gene drives can propagate desirable traits for controlling disease spread into a native population, gene-drive propagation is difficult to control due to the always-active nature of the Cas9 nuclease. To develop some layers of control over the self-propagation of gene drives, multiple strategies have been developed[1]. For example, a split gene drive was built by placing the Cas9 and gRNA components at two different loci into two different transgenic lines; here, the gene drive was activated only when these two strains were combined by genetic crosses[7,8]. Yet, the generation of multiple transgenic lines, especially for organisms with difficulties in obtaining high transgenic efficiencies such as mosquitoes[9,10] could be an issue. To fine-tune gene drives using a small molecule, a drug-regulated system was developed, but this approach required a less efficient modified Cas9 and a more complex genetic circuitry[11,12]. Lastly, ECHACR, ERACR, and anti-CRISPR approaches could also help control gene-drive

[1]Department of Epigenetics & Molecular Carcinogenesis at MD Anderson, The University of Texas MD Anderson Cancer Center, Houston, TX 77054, USA. [2]Center for Cancer Epigenetics, MD Anderson Cancer Center, Houston, TX 77054, USA. [3]Section of Cell and Developmental Biology, University of California San Diego, La Jolla, CA 92093, USA. [4]University of Texas Health Science Center, School of Public Health, Department of Epidemiology, Human Genetics, and Environmental Sciences, Center for Infectious Diseases, Houston, TX 77030, USA. [5]Institute of Infectious Diseases, Shenzhen Bay Laboratory, Shenzhen, Guangdong 518106, China. ✉e-mail: michelle626489@gmail.com; victor.lopezdelamo@uth.tmc.edu

activities[13,14], but these systems may not be preferable as they require a secondary release of genetically modified animals, imposing potential regulatory issues.

Furthermore, current gene-drive methods majorly employ a Cas9 nuclease[1], limiting the creation of arrangements towards next-generation gene-drive systems. Yet, other nucleases, such as Cas12a, have demonstrated efficient genome editing in a broad range of organisms[15–17], including a proof-of-concept gene drive in budding yeast[18]. In *Drosophila melanogaster*, Cas12a-based genome editing produced efficient gene disruption[19,20]; however, the DNA homology-directed repair (HDR) triggered by Cas12a has yet not been explored in insects. Therefore, a functional gene drive system employing Cas12a in flies would offer new opportunities for next-generation genetic strategies using insects (i.e., mosquitoes) to fight vector-borne diseases.

Based on previous works regarding Cas12a-involved genome editing, we hypothesized that the Cas12a-mediated gene drive should be a suitable strategy for addressing the lack of control over Cas9 activity while increasing the available toolkit for future genetic strategies for vector control. First, a Cas12a gene-drive system could be modulated with temperature, which would increase our control over its self-propagation, thereby adding an extra level of containment. In principle, Cas12a gene drives should remain inactive at low temperatures while activating their self-propagation abilities at higher temperatures. Second, a functional Cas12a gene-drive system would be orthogonal to a Cas9 one, which should open new avenues toward developing next-generation arrangements such as double-drive systems, where two gene-drive elements (driven by distinct nucleases) spread concurrently. For these reasons, we wanted to explore the use of Cas12a in a gene-drive setting.

Thus, we herein explored the feasibility of a temperature-regulated Cas12a gene-drive system in *Drosophila melanogaster*. Indeed, we first present a Cas12a gene-drive system that promotes super-Mendelian inheritance in *Drosophila melanogaster*, revealing that Cas12a can trigger efficient HDR in the germline in a temperature-dependent manner. In addition, we also built a double gene-drive system by placing Cas9 and Cas12 together, and demonstrated that two independent genetic elements could spread concurrently within the same organism. This work presents a Cas12a-based gene-drive system in insects while bringing new opportunities for developing alternative gene drives to control disease vectors and agricultural pests.

## Results

### A Cas12a gene-drive system induces super-Mendelian inheritance in a temperature-dependent manner

To determine if the Cas12a nuclease could generate a functional gene drive, we employed a CopyCat gRNA-only gene-drive strategy[11,21] using *Drosophila melanogaster*. Briefly, a CopyCat gene-drive element carries a gRNA gene and two homology arms neighboring the gRNA cut site, allowing the gRNA cassette to copy itself onto the homologous chromosome. The CopyCat transgene propagates only when combined with a Cas nuclease transgene, which itself is inherited in a Mendelian fashion (50%) (Fig. 1a)[11,21].

To test the Cas12a-based gene drives, we built three different transgenic lines: i) one carrying a Cas12a cassette inserted into the *yellow(y)* locus (X chromosome [Chr.X]) and marked with DsRed, here we utilized a point mutation variant of LbCas12a carrying a D156R mutation, which has been previously demonstrated to exhibit high mutagenesis rates in *Drosophila*[20]; ii) a CopyCat transgenic line carrying the *e1*-gRNA under the control of the *Drosophila* U6:3 promoter and *Opie2*-GFP inserted in the *ebony* locus (Chr.III) for tracking the transgene; and iii) another CopyCat transgenic line carrying the *e4*-gRNA under the control of *Drosophila* U6:3 promoter and *Opie2*-GFP inserted in the *ebony* locus (Chr.III) (Supplementary Fig. 1a). It is important to note that our Cas12a-based CopyCat gene drives were built such that both homology arms cover the five–nucleotide-

overhang staggered cuts from the Cas12a nuclease, so the homology arms have five overlapping nucleotides on their respective ends (Supplementary Fig. 1b).

To show the feasibility of the Cas12a-mediated gene drive, we first evaluated our system without temperature control by keeping the flies at the standard incubation temperature of 25 °C. Next, we crossed Cas12a transgenic males to the CopyCat gene-drive transgenic females (either *e1*-gRNA or *e4*-gRNA) (Fig. 1b; F0 generation). Then, we collected F1 transheterozygous females carrying both Cas12a and CopyCat transgenes (Fig. 1b; F1) and performed single-pair crossings to *ebony* mutants (*e-/e-*) to evaluate super-Mendelian inheritance via the GFP marker of the gene-drive element in the F2 progeny (Fig. 1b). If the gene drive is active, the GFP marker should score inheritance rates higher than 50%. Indeed, we observed super-Mendelian inheritance in both cases, with the *e1*-gene drive (*e1*-GD) and the *e4*-GD element showing inheritance rates of 84% and 64%, respectively (Fig. 1c; Source Data file 1). We also scored the inheritance of the Cas12a (marked with DsRed) in both cases and confirmed its expected Mendelian inheritance of 50% (Source Data file 1).

To further investigate whether temperature could regulate the super-Mendelian rates displayed by our gene-drive elements, we followed the same approach as above (Fig. 1b) and repeated the experiments at 18 °C and 29 °C. As we would hope for a system with reduced activity at lower temperatures, we observed reduced super-Mendelian inheritance rates of 75% (*e1*-GD) and 52% (*e4*-GD) at 18 °C (Fig. 1c), and increased inheritance rates of 89% (*e1*-GD) and 81% (*e4*-GD) at 29 °C. Indeed, these results indicate the temperature-dependent behavior of our Cas12a gene-drive system, providing new opportunities to control gene-drive activity (Fig. 1c; Source Data file 1).

As the *ebony* gene is a recessive locus, our experimental design made it possible to obtain estimated conversion rates/HDR, deletion and/or insertions (indels) rates, and wildtype/uncut rates for our gene-drive elements based on phenotypes and GFP presence (Supplementary Fig. 2). At the standard temperature of 25 °C, we observed a conversion rate of 91% and 93% for *e1*- and *e4*-GD, respectively, indicating overall high HDR efficiency of the Cas12a-mediated gene drive. In line with this high conversion rate, we only observed the formation of resistant alleles in 8% (*e1*-GD) and 5% (*e4*-GD) of gene-drive individuals. Lastly, we detected 9% and 29% uncut alleles for our *e1*-GD and *e4*-GD at 25 °C, respectively (Supplementary Fig. 2; Source Data file 1). This indicates that the gene-drive elements do not have 100% cutting rates and suggests that the *e1*-GD is more active than the *e4*-GD at 25 °C, explaining the observed inheritance differences between the gene-drive elements.

To molecularly confirm our estimations based on phenotype (Supplementary Fig. 2), we performed next-generation sequencing of the ebony and wild-type (WT) individuals collected from the F2 progeny of our *e1*-GD and *e4*-GD experiments conducted at 25 °C. The sequencing results revealed a variety of indels within our group of F2 ebony mutant individuals, and the most common modifications are depicted in Fig. 1d. Interestingly, none of these deletions erased the PAM sequence, and the most prevalent mutation for both gene drives was a −12-nucleotide deletion (Fig. 1d; Source Data file 2). Furthermore, we observed that only the *e4*-GD generated smaller mutations, such as −2 and −5 nucleotide deletions (Fig. 1d; Source Data file 2), suggesting indel types generated may be gRNA-dependent. Additionally, we confirmed that no in-frame indels were generated in the F2 individuals displaying WT body color, as they exhibited a similar percentage of WT alleles compared to our control Oregon (OrR) flies at both target sites (Source Data file 2). It is important to note that we considered any modification below 1% of the total reads as baseline noise.

Overall, we observed an increase in super-Mendelian inheritance rates when experiments were performed at higher temperatures with both alleles (*e1*-GD and *e4*-GD). Also, *e4*-GD showed no gene-drive activity at 18 °C, providing a temperature-sensitive option for

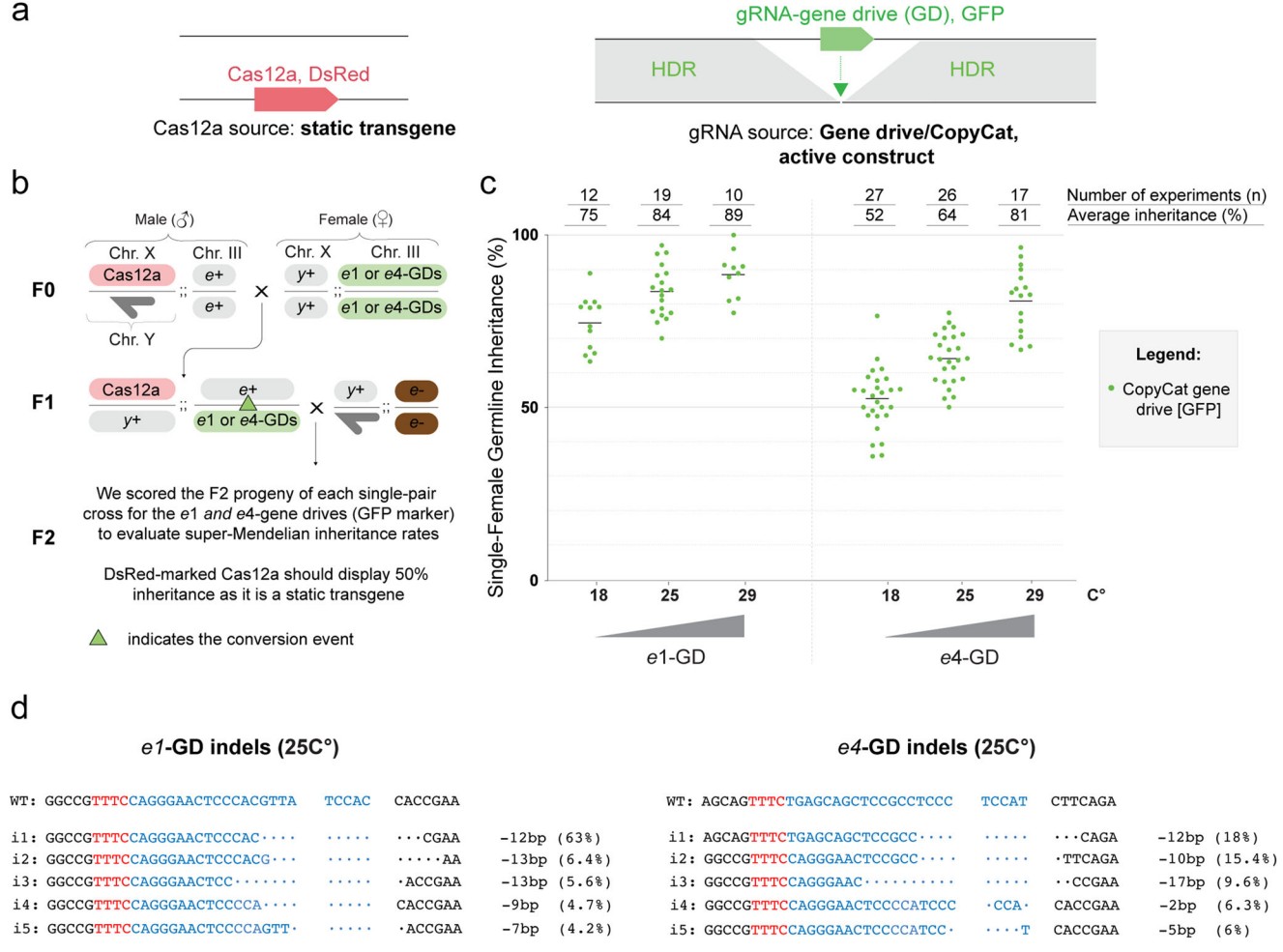

**Fig. 1 | Cas12a-based gene drives display super-Mendelian inheritance rates modulated by temperature. a** Schematic of the CopyCat gene-drive system. The DsRed-marked Cas12a is a static transgene that provides the nuclease for copying the GFP-marked CopyCat element by allelic conversion, which is driven by the surrounding homology arms. **b** Cross scheme of males expressing Cas12a crossed to virgin females carrying the ebony CopyCat constructs (*e1* or *e4* gene drives). Collected F1 virgin females (Cas12a-DsRed + gene drive-GFP) were crossed to ebony mutant males to score germline transmission rates by screening the GFP marker in the F2 progeny. The dark-gray half arrow indicates the male Y chromosome. The green triangle in the F1 female indicates potential gene drive copying onto the wildtype chromosome. **c** Assessment of gene-drive activity in the germline of F1

females by phenotypically scoring the F2 progeny for the GFP-marked ebony CopyCat constructs. Measurements of inheritance rates are reported on top of the graph along with the average inheritance (%) (also as black bars on the graph) and the number of F1 crosses performed (n). **d** Summary of mutations/indels generated by both gene drives tested at 25 °C were detected through deep-sequencing analysis. The most prevalent indels (i) are depicted (i1-i5). Spaces between nucleotides represent the staggered cuts introduced by the Cas12a. The PAM sequence is highlighted in red, and gRNA is highlighted in blue. Nucleotides in black represent the genomic region surrounding the target site. The indel types and their relative percentages are shown next to each sequence. The raw scoring data and statistics are provided in Source Data file 1.

controlling these active genetic elements. Furthermore, our deep-sequencing analysis allowed us to characterize indel formation types, which should provide valuable insights for future studies, as discussed below.

## Cas9 and Cas12a promote the propagation of two independent gene drives within the same organism

To show the feasibility of two independent gene drives spreading simultaneously in a single organism, we employed a double CopyCat gene-drive strategy. The two different Cas nucleases (Cas9 and Cas12a) were inserted individually in different chromosomes as static transgenes: Cas9 was inserted in Chr. III and Cas12a was inserted in Chr. X. (Fig. 2a). For the CopyCat gene-drive elements, we combined the *w2*-gRNA (*w2*-GD) with Cas9 to target the *white* gene in Chr. X.; this gRNA activity was already validated in our previous studies[7,11].

For the second gene-drive element, we combined the *e1*-GD with the Cas12a nuclease to target the *ebony* gene (Fig. 2a); the *e1*-gRNA was validated in our results in Fig. 1. To allow independent tracking when

scoring, we marked all transgenes with various fluorescent markers located in different fly body parts. Cas12a and Cas9 were marked with DsRed in the eye and thorax, respectively, and the gRNA-only gene drives driven by Cas12a (*e1*-GD) and Cas9 (*w2*-GD) were marked with GFP in the abdomen and eye, respectively (Supplementary Fig. 1a).

To assess their performance, we crossed Cas12a/Cas9 females with the gene-drive transgenic males containing the two above mentioned gene-drive elements. Subsequently, we collected transheterozygous F1 females containing all four transgenes and crossed them to *white* and *ebony* mutant males to score transgene inheritance in the F2 progeny (Fig. 2b).

Importantly, the genetic arrangement of our double gene drive allowed us to accurately assess the overall inheritance, HDR, indel and WT allele rates for both gene drives. Here, the static Cas12a element marked with DsRed in the eye is inserted at the *yellow* locus (*y*-Cas12a; chromosome. X) while serving as the receiver chromosome for the *w2*-GD (marked with GFP in the eye). Importantly, the receiver chromosome contains a wildtype *white* gene, which can be targeted for

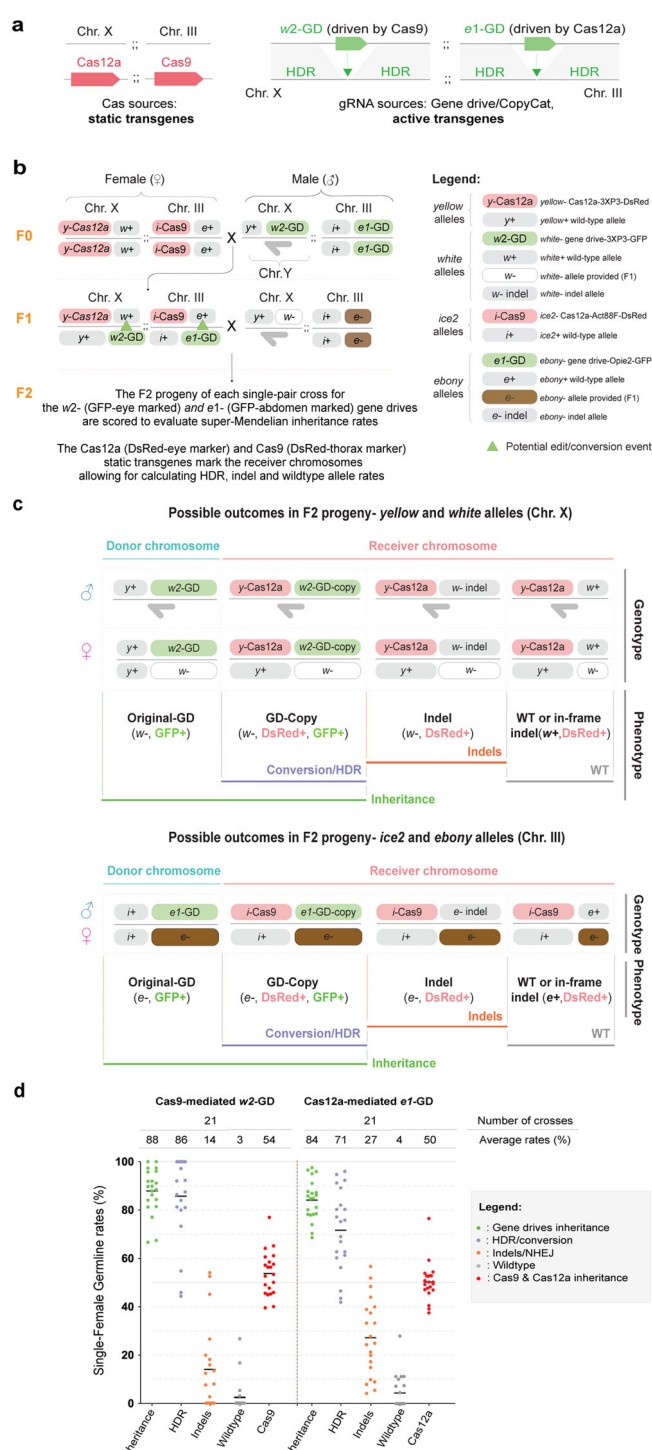

**Fig. 2 | The double gene drive system driven by Cas9 and Cas12a displays simultaneous super-Mendelian inheritance rates of two independent genetic elements. a** Schematic of the double CopyCat gene-drive system. The DsRed-marked Cas12a and Cas9 represent the nuclease sources as static transgenes. Two independent CopyCat gene-drive elements marked with GFP (*w2*-GD and *e1*-GD) are driven by the Cas9 or Cas12a nucleases, respectively. To track their inheritance, we marked all four transgenes without overlapping fluorescent markers in any body parts. **b** Cross scheme of males expressing Cas12a and Cas9 crossed to virgin females carrying two CopyCat constructs (*e1* and *w2* gene drives). Collected F1 virgin females (Cas12a-Cas9-DsRed; gene drives-GFP) were crossed to ebony/ white mutant males to score germline transmission rates by screening the GFP markers in the F2 progeny. The donor chromosomes in *w2*-GD and *e1*-GD were marked by Cas12a and Cas9 transgenes with different fluorescent markers, respectively. A dark-gray half arrow indicates the male Y chromosome. The green triangles in the F1 female indicate potential gene drive copying onto the wildtype allele. **c** Possible outcomes in F2 progeny (Top panel: *w2*-GD in Chromosome. X; Bottom panel: *e1*-GD in Chromosome. III). The outcomes were classified into four categories based on the phenotypic readouts in F2 progenies: 1) Original gene drive from donor chromosome, represented as individuals with *w-* or *e-*; GFP+ phenotypic markers; 2) Allelic conversion or HDR events, represented as individuals with *w-* or *e-*; GFP+ and marked with DsRed+ by Cas elements; 3) Indel/NHEJ events, represented as individuals with *w-* or *e-*; GFP- and marked with DsRed+ by Cas elements; and 4) wildtype represented as *w+* or *e +*; GFP- and marked with DsRed+ by Cas elements. **d** Evaluation of gene drive parameters in the germline of F1 females by phenotypically scoring the F2 progeny. The inheritance, HDR/conversion, indels, and wildtype averages are represented on the top of the graph along with the number of F1 germline scores (n). Each dot represents the data from a single F1 female. Source data is provided as a Source Data File 3.

Additionally, the presence of DsRed-marked eye or DsRed-marked thorax along with the *white* eye or *ebony* body color phenotypes in F2 progeny represent indel events produced by the *w2*-GD and *e1*-GD, respectively (Fig. 2c). Lastly, non-GFP individuals, carrying DsRed-eye or DsRed-thorax markers along with wildtype eye or body color indicate the presence of wildtype alleles where no genome editing events occurred (Fig. 2c).

First, we observed Mendelian inheritance rates (~50%) for both the Cas9 and Cas12a nucleases, as expected (Fig. 2d; Source Data file 3). Also, we found an average inheritance rate of 88% for the *w2*-GD (driven by Cas9) and an average inheritance rate of 84% for the *e1*-GD (driven by Cas12a) in the F2 offspring (Fig. 2d; Source Data file 3). Importantly, our validated *w2*-GD displayed an average HDR rate of 86%, an average indel rate of 14%, and 3% wildtype allele rates (Fig. 2d; Source Data file 3). These results are consistent with our previous work when the *w2*-GD was tested alone[11], suggesting that the Cas9-based gene drive is not affected by the presence of Cas12a. In contrast, the *e1*-GD driven by Cas12a showed an average HDR rate of 71%, 27% indels, and 4% wildtype allele rates (Fig. 2d; Source Data file 3). These values differ from when the *e1*-GD was tested alone (Supplementary Fig. 1b), as the *e1*-GD exhibited an increased indel rate from 8% to 27%, suggesting that the presence of the Cas9 may impact the efficiency of the Cas12a-based gene drive system.

Overall, these results indicate that two independent gene-drive cassettes driven by distinct nucleases can display simultaneous super-Mendelian inheritance of genetically engineered elements in the same organism.

## Discussion

This paper presents a proof-of-concept for a temperature-controlled Cas12a gene drive and demonstrates the feasibility of two gene drives driven by two different nucleases (Cas9 and Cas12a) propagating in a single organism. We showed that Cas12a could induce super-Mendelian inheritance in two different genetically engineered alleles (*e1*-GD and *e4*-GD) in a temperature-dependent manner. Both gene drives, driven by validated gRNAs targeting the *ebony* gene for disruption[20], displayed the desirably lower super-Mendelian rates at

conversion by the *w2*-GD from the donor chromosome in transheterozygous F1 females (Fig. 2b, c). Since the *yellow* and *white* loci are linked (they are inherited together), HDR events occurring on the *y-*Cas12a receiver chromosome are represented by F2 individuals displaying both the DsRed-eye and GFP-eye marker (Fig. 2c). A similar approach is employed for the *e1*-GD driven by Cas12a. In this case, Cas9 is inserted at the *ice2* locus (*i-*Cas9; chromosome. III), which is linked with the *ebony* locus. Here, the *i-*Cas9 marked with DsRed-thorax represents the receiver chromosome, where the *e1*-GD marked with GFP in the abdomen (donor chromosome) can target the wildtype allele of the *ebony* gene (Fig. 2b, c). Thus, the coexistence of the DsRed-thorax and GFP-abdomen markers in F2 individuals indicates copying events (HDR) occurring on the receiver chromosome (Fig. 2c).

18 °C and increased gene-drive efficiency at 25 °C and 29 °C. Our results demonstrate that Cas12a-based HDR, which has been extensively used in other organisms[17,18,22], can also be achieved in a gene drive context using *Drosophila melanogaster*. Indeed, this work shows the functionality of a Cas12a gene-drive system triggering super-Mendelian inheritance rates in an insect germline. Interestingly and in line with the activity of Cas12a at lower temperatures, we did not observe super-Mendelian inheritance rates when the *e4*-GD experiments were performed at 18 °C, indicating that Cas12a-based gene-drive inactivation may be possible by switching to lower temperatures. However, it is important to mention that the *e1*-GD still displayed super-Mendelian inheritance rates at 18 °C. This observation could be explained by differences in gRNA activities as distinct gRNAs promoted variable levels of disruption when targeting the *ebony* gene[20].

Altogether, our experiments show that the use of a Cas12a nuclease can provide an additional layer of temperature-dependent containment to regulate gene-drive propagation under laboratory conditions. In addition to providing some level of inactivity when desired, unlike to the always-on Cas9 nuclease, gene-drive containment is an important feature demanded by the gene-drive community and regulators[23–26]. Also, these Cas12a-based gene-drive systems displayed super-Mendelian inheritance rates comparable to previous Cas9-based systems that efficiently spread in a target population under laboratory conditions[27]. Our previous work described a gene-drive system in *Drosophila* showing ~85% super-Mendelian rates targeting the *ebony* gene while using a Cas9 cassette inserted into the *yellow* locus and driven by the *vasa* promoter[7], which is comparable to the inheritance rates observed in this work with *e1*-GD driven by the Cas12a, which was also inserted into the *yellow* locus and driven by the same *vasa* promoter. However, future investigations are needed to explore the propagation dynamics of Cas12a-based systems by performing caged population studies.

The formation of resistant alleles is also a concern for gene-drive propagation[28,29]. Interestingly, Cas12a introduces DNA breaks, which are far away from the DNA recognition site (PAM) compared to Cas9. In fact, Cas12a has been observed to introduce small indels that are far away from the PAM sites in plants and zebrafish[17,30]. In this study, we observed that deletions generated by our Cas12a-based gene drives retained the PAM sequence. Specially, the most prevalent mutation is a 12-nucleotide deletion in the target region. Furthermore, the *e4*-GD displayed smaller deletions (−2nt and −5nt) in the PAM-distal region (Fig. 1d). Since the PAM proximal seed region of the gRNA is crucial for DNA recognition and targeting[31], the small PAM-distal mutations generated by our system could still be recognized and targeted by the Cas12a-based gene drive, thereby improving its propagation and enabling its full introgression into a target population. Future studies should target different genes to investigate their ability to generate small indels distal to the PAM site while broadening potential targets for Cas12a next-generation gene-drive designs. Indeed, these observations deserve further investigation to mitigate the burden of resistant alleles imposed by Cas9-based systems in certain situations[28,29,32–34].

In addition, a Cas12a system should improve current Cas9-based genetic sterile insect technique (gSIT) that involves high maintenance costs from rearing two separated stocks (Cas9 and gRNAs) and sexing of the separated strains to generate sterile males through genetic crosses[35,36]. By using a Cas12a nuclease instead, a single strain containing Cas12a and gRNAs together could be generated and kept as a regular stock at 18 °C, since Cas12a should be inactive. Then, sterile males would be generated by switching the stock to higher temperatures[37] while avoiding the crossing step required in Cas9 settings. This could potentially reduce logistical issues when high throughput rearing is needed (i.e., for field release purposes).

Once we demonstrated the feasibility of a gene-drive approach using Cas12a, we also questioned whether the propagation of two independent gene-drive elements driven by two different nucleases is possible in the same organism. The combination of two transgenic lines, one containing Cas9 and Cas12a and a second line containing two gRNA-only drive elements driven by each of the nucleases, showed super-Mendelian inheritance rates for *w2*-GD (88%) and *e1*-GD (84%) at 25 °C. Both gene-drive elements showed comparable inheritance rates when tested singularly. *e1*-GD had same inheritance rates: 84% singularly and when combined with the *w2*-GD. We previously tested the singular *w2*-GD driven by Cas9 and detected 90% super-Mendelian inheritance rates[11], which is similar to the 88% inheritance rate observed in our double gene-drive system. These small differences in inheritance rates are most likely due to the different genomic locations of Cas9 and/or the promoter employed in each situation, as we previously reported[7]. In this work, we utilized a Cas9 located in the third chromosome driven by the *nanos* promoter, whereas when *w2*-GD was tested singularly before, it was driven by a Cas9 inserted in the X chromosome driven by the *vasa* promoter[7].

Regardless, this double gene-drive system promoting super-Mendelian inheritance rates of two independent transgenes opens an interesting avenue for designing innovative approaches with two distinct nucleases. For example, this could be implemented to design sex ratio distortion systems for population suppression that require simultaneous homing in pre-meiotic cells and chromosome shredding at the early zygote stage[38,39]. Here, Cas9 and Cas12a could be driven by specific promoters to ensure their expression at the particular tissues. Also, we could design new insect-vector population suppression strategies such as the proposed double drives, which target two different genomic locations[40]. A double drive system, in which Cas9 and Cas12a would target sex-determination genes at two locations (within the same gene or at distinct loci), could improve gene drive spread, as indels generated at one target site would not affect the propagation of the other gene drive cassette, and ultimately ensure its full propagation while improving current suppression strategies. It is important to note that Cas9 and Cas12a have different PAM requirements and should not interfere with each other; Cas9 needs a NGG PAM site and Cas12a requires a TTTN sequence for targeting[41]. Therefore, the use of Cas12a for vector control will also facilitate targeting genomic regions that we could not attack before due to restrictions with the Cas9 PAM site.

Overall, our work will provide relevant information to rapidly deploy Cas12-based strategies in disease-relevant insects. Furthermore, future applications of Cas12a-based gene drives should be applicable to different organisms, as evidenced by the plethora of CRISPR gene drives that have been successfully applied to other systems, including mice, bacteria, and viruses[42–45]. Altogether, our proof-of-concept studies provide a platform that will promote further development of Cas12a-based gene-drive systems for improved insect-vector population control strategies and encourage its implementation in a wide range of organisms.

## Methods

### Ethics

The experiments in this work adhere to all relevant ethical regulations and follow the approved protocols and procedures of the UTHealth Institutional Biosafety Committee at UTHealth Science Center Houston.

### Plasmid construction

All plasmids were built following standard molecular biology approaches. Plasmids were constructed by Gibson Assembly using NEBuilder HiFi DNA Assembly Master Mix (New England BioLabs Cat. #E2621) and transformed into NEB 10-beta electrocompetent *E.coli* (New England BioLabs Cat. #3020). Plasmids were purified using a Qiagen Plasmid Midi Kit (Qiagen Cat. #12143), and DNA sequences were confirmed by Sanger sequencing. All constructions and their NCBI ID were shown in Supplementary Fig. 1.

## Generation of transgenic lines

Plasmids of Cas nucleases and the gene drive (CopyCat) were sent to Rainbow Transgenic Flies, Inc. for injection with a Cas9/Cas12a-expressing plasmid. For Cas insertions, we injected a plasmid containing the Cas nucleases and homology arms for integration along with a second plasmid expressing a gRNA targeting *yellow* for Cas12a and inserted it into the *ice2* gene in linkage with the ebony locus for Cas9 integration. CopyCat gene-drive constructs were injected into transgenic flies already expressing Cas9 or Cas12a to facilitate CopyCat insertion. The injected $G_0$ larvae were sent back to us and were allowed to develop. The adult flies hatching from injected embryos were crossed to each other in pools of 3–5 males with 3–5 females. The resulting $G_1$ individuals were then screened for green fluorescence in their abdomen as a marker of the CopyCat transgene insertion (*e1*- and *e4*-GDs). Cas12a insertion was identified by scoring DsRed fluorescence in the eye. Cas9 transformants were identified by scoring DsRed in the thorax. Homozygous lines for each strain were constructed from single transformants crossed to flies of the opposite sex and following the recessive body color phenotype and/or fluorescent marker in subsequent generations. Proper transgene integration in each strain was molecularly confirmed by PCR and Sanger sequencing.

## Statistics and reproducibility

Flies were fed on standard cornmeal. Fly stocks were kept at 18 °C, and all experimental crosses and cages were maintained at 18 °C, 25 °C, or 29 °C, depending on the experiment. Flies were anesthetized with $CO_2$ when scoring phenotypes and preparing crosses. For all experimental crosses, virgin females were collected as pupae and crossed on the same day they were hatched. $F_0$ crosses were made in pools of 3–5 males crossed to 3–5 females. $F_1$ females were single pair crossed to wild-type males and left for five days before removing the flies (Figs. 1 and 2). After all the flies emerged, $F_2$ flies were scored for sex, body/eye color, and fluorescence (DsRed and GFP) as a marker of transgene inheritance using a Leica M165 F2 stereomicroscope with fluorescence. The experiments were randomized. Experimental crosses where contamination impeded proper fly development were excluded from the analyses. All gene drive experiments were performed in a high-security ACL2 (Arthropod Containment Level 2) facility. Crosses were carried out in shatter-proof polypropylene vials (Genesee Scientific Cat. #32-120). All flies and vials were frozen for 48 h before being removed from the facility, autoclaved, and discarded as biohazardous waste. All graphs were generated using GraphPad Prism 9 and Adobe Illustrator. Statistical analyses were performed using GraphPad Prism 9. To analyze differences between *e1*-GD and *e4*-GD tested at different temperatures, two-way ANOVA and Tukey's multiple comparisons test were performed (Fig. 1).

## Deep-sequencing analysis

To perform deep-sequencing of Cas12a-mediated GD experiments in Fig. 1, we collected GFP- individuals in the F2 progeny with either an *ebony* phenotype (representing potential NHEJ/indel events) or wildtype phenotype (representing wildtype/unedited alleles), from both *e1* and e4 *GD*s' experiments. The control OregonR (OrR) flies were used as a baseline control. We pooled flies derived from at least 10 independent germline crosses for each condition. For the *e1* target, F2 individuals with GFP-; ebony phenotype were pooled for tracking indels ($n = 30$); F2 individuals with GFP-; wildtype body color were pooled for tracking wildtype alleles ($n = 30$); and OrR individuals that represent our baseline control ($n = 20$). For the *e4* target, F2 individuals with GFP-;ebony phenotypes were pooled for tracking indels ($n = 18$); F2 individuals with GFP-; wildtype body color were pooled for tracking wildtype alleles ($n = 49$); The OrR individuals ($n = 20$) were served as controls. The deep sequencing results can be found as Source data provided as a Source Data File 2. The genomic

DNA was extracted from each fly pool following the standard protocol in our previous study[9]. After extraction, about ~500 ng of the extracted DNA was then used in a 50 uL PCR reaction as a template to amplify either the *e1* or *e4* targeted region using specific primers with adapters (Primers are listed in the Supplementary Methods). PCR products were purified in agarose gel using a QIAquick Gel Extraction Kit (Qiagen). The purified products were sent to Azenta/Genewiz company for deep-sequencing analysis. Occurrences with less than 1% of reads were considered background noise. Primers used for DNA amplification can be found in Source Data file 4.

## Reporting summary

Further information on research design is available in the Nature Portfolio Reporting Summary linked to this article.

## Data availability

The plasmids generated in this study have been deposited in the GenBank database (https://www.ncbi.nlm.nih.gov/genbank/) with the following accession numbers: OR262462,OR262463, OR262464, OR262465, and OR262466. In addition, all Supplementary Information covering the raw phenotypical scoring data collected in the gene-drive experiments is provided in Microsoft Excel format (.xlsx) as Source Data files, as indicated in the manuscript. For other resources such as transgenic flies, they are available upon request after completing the material transfer agreement (MTA). Source data are provided with this paper.

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

## Acknowledgements

We thank Kaycie Butler and Ahana Maitra for their comments and edits on the manuscript. We thank Valentino Gantz for his scientific feedback. We thank Fillip Port for kindly providing the Cas12a plasmid. The research reported in this paper was supported by the Rising STARs Program awarded by The University of Texas System Board of Regents (to V.L.D.A), and by the National Natural Science Foundation of China grant 82202559 (to X.F).

## Author contributions

V.L.D.A and X.F conceived the project. V.L.D.A and X.F designed and obtained the plasmids and Drosophila transgenic lines. V.L.D.A, X.F, S.S.J, E.O and C.N performed the experiments. V.L.D.A, X.F and S.S.J performed the figure visualizations. V.L.D.A wrote the manuscript, which was edited by all the authors.

## Competing interests

V.L.D.A and X.F. are authors on a patent filed by the University of California, San Diego (PCT/US2022/044487) related to the Cas12a gene-drive system described in this work. This patent has been published and more information can be found in Google Patents (https://patents.google.com) under the accession number WO2023064084A1. All authors declare no competing interests.
