## [Peer Review File · Nature Communications]

Reviewers' Comments:

Reviewer #1:

Remarks to the Author:

This paper describes experiments investigating the potential use of Cas12a nuclease for gene drive constructs. All previous work on this important and timely topic has been with Cas9, and Cas12a differs in some important ways, including the nature of the ends left after cleavage, which could affect homing rates.

Unfortunately, the results are a bit thin. One interesting question is whether there is any difference in homing rates between the two enzymes, but no comparison is possible because they are in different locations, driven by different promoters, and cleave sequences in different genes. Based on a single Cas12a insertion, with a single promoter, and 2 different cleavage sites, all we can conclude is that the rates may be comparable.

Another interesting question is how the presence and activity of one enzyme affects the activity of the other, which could have been investigated in a single experiment with all combinations of constructs, but instead we have to compare across experiments, or even across studies, where many other differences exist, such as genomic location and promoter used. So, all we can conclude is that they can work together — and even this we only know for the case where they have different promoters, and so may be active in somewhat different cell populations.

Methods: need to make very clear which Cas12a was used.

Lines 102-5 — I do not understand this sentence or the figure — please can you make more clear?

Discussion — what is meant by 'containment' — is this to prevent escape from the lab? Or to prevent geographical spread after a release?

Lines 255-7 — perhaps worth noting that having 2 different nucleases would be particularly useful for combined strategies that involve both homing in pre-meiotic cells and chromosome drive like X-shredding in meiosis (or any other case where it is important for the 2 drivers to act in different tissues).

Reviewer #2:

Remarks to the Author:

This manuscript presents and validates the concept that Cas12a-mediated gene drives could be regulated by temperature, for the first time demonstrating the feasibility of simultaneous propagation of two independent gene-drive elements mediated by two different nucleases within the same organism. These findings expand the toolkit selection for future genetically engineered vector and pest control strategies, whereas Cas12a-based gene drives systems and multi-loci CRISPR gene drive concepts have been published previously. I think the work is sound, but the novelty is moderate.

Here are some specific comments:

1. In the abstract, the authors emphasized that Cas12a has not been explored for gene drive designs, but the Cas12a-mediated gene drive in budding yeast was reported previously (Access Microbiology, PMID:35024561). The authors should rewrite related sentences.
2. In this paper, the authors tested the Mendelian heritability of static transgenes and active constructs at three laboratory incubation temperatures: 18°C, 25°C, and 29°C, finding that temperature could regulate the Cas12a-based CopyCat gene-drive activity. But compared with the e4-GD, the e1-GD displays weak sensitivity relative to temperature. Temperature-sensitive regulation of gene drive is the main innovative point in the manuscript, which should be enhanced. For example, the authors may conduct a screening to identify more highly temperature-sensitive target sites at drivable genes.

3. In lines 142-144 and Supplementary Figure 2, the authors discuss the possibility of estimating the conversion rates/HDR, the formation rates for indels/resistant alleles, and wildtype/uncut rates by three phenotypes: GFP+, ebony-, and ebony+, of which GFP+ represents the gene conversion rates, ebony- represents the indels allele rates. However, the GFP+ phenotype in F2 includes two parts in principle: the ebony- Gene drive-GFP transgene and converted GFP transgene which originally was the ebony+ gene. The authors need to explain clearly how they calculated the HDR rate. Moreover, it may still show an ebony-positive phenotype if the ebony+ wildtype allele undergoes NHEJ repair accompanied by in-frame indel, so the real indel rate needs to be defined by other assays such as target sequencing.

4. The multi-loci CRISPR gene drive with single Cas9 has been reported (Scientific Reports, PMID:30467400). What are the advantages of two independent gene drives mediated by two different nucleases compared with using a single Cas9 or Cas12a to combine two gRNAs to achieve a dual gene drive?

5. Figure 1C: There are no statistical comparisons for inheritance between the different temperatures. The authors should add the statistical analysis.

6. Is there a direct correlation between the CopyCat gene editing efficiency and the ultimate gene-driving ability? Also, is there an editing efficiency threshold below which there is no gene-driven ability at 18°C?

REVIEWER COMMENTS

We sincerely appreciate the valuable feedback provided by both reviewers on our manuscript. The reviewers' comments helped us improve the manuscript substantially, and we hope to address all concerns raised and answer the questions satisfactorily. We are providing a revised version of the manuscript, including the suggested revisions and edits, with comments highlighting the modifications that address each specific comment. Please find point-by-point responses written in blue text below.

Reviewer #1 (Remarks to the Author):

This paper describes experiments investigating the potential use of Cas12a nuclease for gene drive constructs. All previous work on this important and timely topic has been with Cas9, and Cas12a differs in some important ways, including the nature of the ends left after cleavage, which could affect homing rates.

1. Unfortunately, the results are a bit thin. One interesting question is whether there is any difference in homing rates between the two enzymes, but no comparison is possible because they are in different locations, driven by different promoters, and cleave sequences in different genes. Based on a single Cas12a insertion, with a single promoter, and 2 different cleavage sites, all we can conclude is that the rates may be comparable.

We appreciate the comment from the reviewer. Regarding the comparison between Cas9 and Cas12a in a gene drive context, our previous study (Lopez del Amo, 2020, *Nature communications*) described a gene-drive system in *Drosophila*, targeting the *ebony* gene, which achieved approximately 85% inheritance rates using a Cas9 nuclease inserted into the *yellow* locus and driven by the *vasa* promoter. In this current work, we utilize Cas12a also inserted into the *yellow* (same location) and driven by the same *vasa* promoter and targeting the *ebony* gene (not the same location, though). The observed inheritance rates in this study were comparable to that achieved by Cas9 in our previous work (~85%). To highlight this point, we have modified the discussion accordingly.

“Our previous work described a gene-drive system in Drosophila showing ~85% super-Mendelian rates targeting the ebony gene while using a Cas9 cassette inserted into the yellow locus and driven by the vasa promoter⁷, which is comparable to the inheritance rates observed in this work with e1-GD driven by the Cas12a, which was also inserted into the yellow locus and driven by the same vasa promoter.”

It is important to note that comparing Cas9 and Cas12a is challenging due to their different PAM requirements and gRNA properties. Therefore, generating a gene drive targeting the same genomic region with both nucleases is technically complex, which is why we chose not to emphasize this aspect. Yet, our proof-of-concept Cas12a system demonstrates super-Mendelian inheritance rates like Cas9, suggesting the potential of Cas12a as an alternative for vector control strategies, as mentioned above.

2. Another interesting question is how the presence and activity of one enzyme affects the activity of the other, which could have been investigated in a single experiment with all combinations of constructs, but instead we have to compare across experiments, or even across studies, where many other differences exist, such as genomic location and promoter used. So, all we can conclude is that they can work together — and even this we only know for the case where they have different promoters, and so may be active in somewhat different cell populations.

As mentioned by the reviewer, our double gene drive system involves the insertion of Cas9 and Cas12a nucleases at different loci, along with *e1*-GD and *w2*-GD targeting two different loci. It is important to note though that the *e1*-GD and Cas12a transgenes are the same in **Fig.1** and **Fig.2**. So, we can investigate how the Cas9 presence may impact Cas12a-based gene-drive activity through our double gene-drive system.

Most important, our double gene drive design allows us to distinguish between donor and receiver chromosomes (see updated **Fig. 2**) while tracking HDR and indels occurring in the F1 individuals by phenotypically scoring the F2 progeny. Please, see explanation below.

In this double gene-drive system, Cas12a is inserted at the *yellow* locus (Chromosome X), *e1*-GD (specific to Cas12a) is inserted at the *ebony* locus (Chromosome III), Cas9 is inserted at the *ice2* locus (Chromosome III), and *w2*-GD (specific to Cas9) is inserted at the *white* locus (Chromosome X). For the *w2*-GD, the Chromosome X containing the Cas12a (marked with DsRed-eye) represents the receiver chromosome, which contains wildtype alleles of the *white* gene. As mentioned, this receiver chromosome could be targeted by the *w2*-GD element (GFP-eye marked) from the donor chromosome (original GD chromosome). Since *yellow* (Cas12a-DsRed) and *white* (*w2*-GD-GFP) are genetically linked (inherited together), the presence of both DsRed-eyes and GFP-eyes in the F2 progeny indicates conversion (HDR) events happening in the receiver chromosome (see updated **Fig.2**).

A similar strategy has been followed by Cas12a driving *e1*-GD. In this case, the receiver chromosome III is marked with the Cas9 element (inserted in the *ice2* locus with a DsRed-thorax marker), which contains wildtype alleles of the *ebony* gene. Then, the receiver chromosome can be targeted by the *e1*-GD (donor chromosome), which is marked with GFP-abdomen. As the *ice2* and *ebony* genes are also genetically linked, the coexistence of DsRed-thorax and GFP-abdomen in the F2 progeny indicates conversion (HDR) events happening in the receiver chromosome. Furthermore, we evaluated mutations/indels, and wildtype/uncut allele rates arising from our double gene-drive system in the F2 progeny. We have modified **Fig.2** and the results section to highlight all these observations.

Regarding the interactions between these two nucleases, we observed that indels generated by the *e1*-GD was 27% when combined with Cas9, which is higher than the 8% observed when Cas12a was tested alone (compare indel rates from **updated Fig. 2d** and **Supplementary Fig. 2c**). This finding suggests potential interactions between these two endonucleases. However, further research is required to explore this phenomenon. We have included these observations in the new resubmission as well.

3. Methods: need to make very clear which Cas12a was used.

We appreciate the reviewer for bringing up this comment. We have included the relevant information in the results section to make it more easily identifiable.

"[...], we utilized a point mutation variant of LbCas12a carrying a D156R mutation, which has been previously demonstrated to exhibit high mutagenesis rates in Drosophila²⁰"

4. Lines 102-5 — I do not understand this sentence or the figure — please can you make more clear?

Due to the staggered cutting sites generated by the Cas12a nuclease, 5-nucleotide overhangs were added on the 5' end of each homology arm when designing the Cas12a-based CopyCat gene drives.

We have updated our **Supplementary Fig.1b** to clarify this point.

5. Discussion — what is meant by ‘containment’ — is this to prevent escape from the lab? Or to prevent geographical spread after a release?

By using the term “containment”, we intended to convey the idea of controlling gene drive activity specifically in laboratory settings. To address this, we have modified the section from:

“Altogether, our experiments show that the use of a Cas12a nuclease can add a temperature-dependent layer of containment over gene-drive propagation”

To:

*“Altogether, our experiments show that the use of a Cas12a nuclease can provide an additional layer of temperature-dependent containment to regulate gene-drive propagation **under laboratory conditions**”*

6. Lines 255-7 — perhaps worth noting that having 2 different nucleases would be particularly useful for combined strategies that involve both homing in pre-meiotic cells and chromosome drive like X-shredding in meiosis (or any other case where it is important for the 2 drivers to act in different tissues).

Thanks for this suggestion. We have added this in the discussion.

“For example, this could be implemented to design sex ratio distortion systems for population suppression that require simultaneous homing in pre-meiotic cells and chromosome shredding at the early zygote stage^{38,39}. Here, Cas9 and Cas12a could be driven by specific promoters to ensure their expression at the particular tissues.”

Reviewer #2 (Remarks to the Author):

This manuscript presents and validates the concept that Cas12a-mediated gene drives could be regulated by temperature, for the first time demonstrating the feasibility of simultaneous propagation of two independent gene-drive elements mediated by two different nucleases within the same organism. These findings expand the toolkit selection for future genetically engineered vector and pest control strategies, whereas Cas12a-based gene drives systems and multi-loci CRISPR gene drive concepts have been published previously. I think the work is sound, but the novelty is moderate.

Here are some specific comments:

1. In the abstract, the authors emphasized that Cas12a has not been explored for gene drive designs, but the Cas12a-mediated gene drive in budding yeast was reported previously (Access Microbiology, PMID:35024561). The authors should rewrite related sentences.

Although we have cited this reference in our previous version, we appreciate this comment and apologize for the oversight in our previous statement. Considering this information, we have revised the relative sentences to clarify them.

“Yet, other nucleases such as the temperature-sensitive Cas12a have not been explored for gene drive designs”

To:

“Yet, other nucleases such as the temperature-sensitive Cas12a have not been explored for gene drive designs in insects”

Also, we have modified our introduction to make it more explicit, from:

“Yet, other nucleases, such as Cas12a, have demonstrated efficient genome editing in a broad range of organisms^{15–18}”.

To:

*“Yet, other nucleases, such as Cas12a, have demonstrated efficient genome editing in a broad range of organisms^{15–17}, **including a proof-of-concept gene drive in budding yeast¹⁸**”*

2. In this paper, the authors tested the Mendelian heritability of static transgenes and active constructs at three laboratory incubation temperatures: 18°C, 25°C, and 29°C, finding that temperature could regulate the Cas12a-based CopyCat gene-drive activity. But compared with the *e4*-GD, the *e1*-GD displays weak sensitivity relative to temperature. Temperature-sensitive regulation of gene drive is the main innovative point in the manuscript, which should be enhanced. For example, the authors may conduct a screening to identify more highly temperature-sensitive target sites at drivable genes.

We agree that the temperature-sensitive regulation of the gene drive is one of the key innovative aspects of our manuscript. As we discussed in our manuscript, the efficiencies of Cas12a's gRNAs can vary, as a previous study has reported different disruption efficiencies targeting the *ebony* gene (ref. 20).

As pointed out by the reviewer, the *e1*-GD exhibited a less sensitivity, as it still showed super-Mendelian inheritance rates to some extent (~60%) at a lower temperature (18°C). In line with this, previous study has shown that *e1*-gRNA displays the highest editing efficiency among multiple gRNAs targeting the *ebony* gene (ref. 20). Therefore, the relatively lower sensitivity of *e1*-GD compared to *e4*-GD is likely due to the higher activity exhibited by the *e1*-gRNA. Additionally, our analysis in **Supplementary Fig.2c** shows that under the same temperature conditions (25°C), the *e4*-GD produces more wildtype alleles compared to the *e1*-GD, indicating that *e4*-gRNA is indeed less active than the *e1*-gRNA.

While it would be interesting to explore other target sites at drivable genes, we plan to pursue these experiments in the future since the primary scope of this manuscript is to show the feasibility of a Cas12a-based gene drive, and expand our toolkit for vector control strategies.

3. In lines 142-144 and Supplementary Figure 2, the authors discuss the possibility of estimating the conversion rates/HDR, the formation rates for indels/resistant alleles, and wildtype/uncut rates by three phenotypes: GFP+, ebony-, and ebony+, of which GFP+ represents the gene conversion rates, ebony- represents the indels allele rates. However, the GFP+ phenotype in F2 includes two parts in principle: the ebony- Gene drive-GFP transgene and converted GFP transgene which originally was the ebony+ gene. The authors need to explain clearly how they calculated the HDR rate. Moreover, it may still show an ebony-positive phenotype if the ebony+ wildtype allele undergoes NHEJ repair accompanied by in-frame indel, so the real indel rate needs to be defined by other assays such as target sequencing.

We agree that we are not able to distinguish the individuals carrying the original gene-drive allele (from the F1 females) and the converted ones (the wildtype allele targeted by the e-GD) in our experiments in **Fig.1** and **Supplementary Fig.2**. This is the reason why we used the “estimated” term in our first submission. To clarify this point, we have updated **Supplementary Figure.2**. Instead, our double gene drive design allowed us to track exact HDR rates. Please, see Reviewer#1 Comment#2, and the **updated Fig.2**.

We also agree with the reviewer's concern regarding potential in-frame indels that could be presented as wildtype alleles. That is also why we depicted this category as WT/in-frame indels in **Supplementary Fig. 2**.

To better address this issue, we followed the reviewer's suggestion, and performed the deep-sequencing analysis in F2 progeny. Specifically, we collected F2 individuals categorized as: 1) GFP-, ebony phenotype representing potential indel events, and 2) GFP-, wildtype phenotype representing wildtype/in-frame indels for both e1-GD and e4-GD conducted at 25°C. We also used Oregon-R flies that served as our baseline control. This analysis showed different indels within the GFP-, ebony phenotype individuals, depicted in **updated Fig.1d** and **Supplementary Data 2**. No in-frame indels at the target site were detected within the GFP-, wildtype body color F2 individuals for both e1 and e4 GDs (see **Supplementary Data 2**). We considered DNA modifications below 1% of the total reads as background noise. We have included all these observations and discussed them in our revised version.

4. The multi-loci CRISPR gene drive with single Cas9 has been reported (Scientific Reports, PMID:30467400). What are the advantages of two independent gene drives mediated by two different nucleases compared with using a single Cas9 or Cas12a to combine two gRNAs to achieve a dual gene drive?

The double gene drive system offers important advantages by incorporating both nucleases instead of using only one. The introduction of Cas12a allows for targeting T-rich regions that are not accessible with Cas9 alone. With this system, one can simultaneously target a gene with an available NGG PAM site using Cas9, and another gene (or other regions within the same gene) that is not suitable for an NGG PAM but instead possesses a TTTN PAM region, which is specific to Cas12a, and significantly expands our DNA target regions, as we mentioned in our previous submission.

Furthermore, another potential advantage of this system that deserves further investigation is the generation of resistant alleles. From our deep sequencing results, which have been included within **Fig.1d** and **Supplementary Data 2** in the revised version, we observed that the *e1*-GD and *e4*-GD generated indels that do not eliminate the PAM site. Consequently, these indels could still be susceptible to the gene drive element in a hypothetical field scenario and could enable the spread of these engineered alleles. In contrast, indels generated by Cas9 usually remove the PAM site, resulting in resistant alleles formation that the gene drive element cannot further target. To highlight this point, we have modified the discussion accordingly:

“In this study, we observed that deletions generated by our Cas12a-based gene drives retained the PAM sequence. Specially, the most prevalent mutation is a 12-nucleotide deletion in the target region. Furthermore, the e4-GD displayed smaller deletions (-2nt and -5nt) in the PAM-distal region (Fig. 1d). Since the PAM proximal seed region of the gRNA is crucial for DNA recognition and targeting³¹, the small PAM-distal mutations generated by our system could still be recognized and targeted by the Cas12a-based gene drive, thereby improving its propagation and enabling its full introgression into a target population. Future studies should target different genes to investigate their ability to generate small indels distal to the PAM site while broadening potential targets for Cas12a next-generation gene-drive designs. Indeed, these observations deserve further investigation to mitigate the burden of resistant alleles imposed by Cas9-based systems in certain situations^{28,29,32–34}.

5. Figure 1C: There are no statistical comparisons for inheritance between the different temperatures. The authors should add the statistical analysis.

We performed statistical analysis comparing all gene drive conditions in **Fig.1**, which can be found in **Supplementary Data 1- Tab 7 and Tab 8**. It is possible an error may have occurred during the pdf. conversion: we have communicated with the editor to make sure the raw data is available for the reviewers.

6. Is there a direct correlation between the CopyCat gene editing efficiency and the ultimate gene-driving ability? Also, is there an editing efficiency threshold below which there is no gene-driven ability at 18°C?

Yes, there is a direct correlation between the CopyCat gene editing efficiency and the ultimate gene drive efficiency. Indeed, a decreased percentage of wildtype (WT) alleles (unedited), represented by the wildtype body color displayed in F2 individuals, was observed in both the *e1*-GD and *e4*-GD as the temperature increased. This trend indicates that the cutting efficiency is higher as the temperature increased; these higher cutting rates also correlate with the increased inheritance observed in both gene drives tested, as illustrated in **Fig.1** and **Supplementary Data 1**.

Regarding the second question, we are unable to conclude any exact editing efficiency threshold below which there is no gene-driven conversion with the two gene drives tested. We can confirm that the *e1*-GD exhibits much higher activity compared to the *e4*-GD. At 18°C, the *e1*-GD still showed (low) editing and gene-drive activity, whereas *e4*-GD appears to be almost inactive (0% editing and therefore 0% HDR). Please, see **Supplementary Data 1** for more information regarding indel and wildtype allele rates.

Reviewers' Comments:

Reviewer #1:

Remarks to the Author:

This is a significantly improved manuscript. I particularly appreciate the addition of the sequence data on the cleavage products, and the additions to the text are also good.

Reviewer #2:

Remarks to the Author:

The authors have addressed my concerns.

We thank both reviewers for appreciating our revised manuscript.